# Luring of transferable adversarial perturbations in the black-box paradigm

## Abstract

The growing interest for adversarial examples, i.e. maliciously modified examples which fool a classifier, has resulted in many defenses intended to detect them, render them inoffensive or make the model more robust against them. In this paper, we pave the way towards a new approach to improve the robustness of a model against black-box transfer attacks. A removable additional neural network is included in the target model, and is designed to induce the *luring effect*, which tricks the adversary into choosing false directions to fool the target model. Training the additional model is achieved thanks to a loss function acting on the logits sequence order. Our deception-based method only needs to have access to the predictions of the target model and does not require a labeled data set. We explain the luring effect thanks to the notion of robust and non-robust useful features and perform experiments on MNIST, SVHN and CIFAR10 to characterize and evaluate this phenomenon. Additionally, we discuss two simple prediction schemes, and verify experimentally that our approach can be used as a defense to efficiently thwart an adversary using state-of-the-art attacks and allowed to perform large perturbations.

## 1 Introduction

Neural networks based systems have been shown to be vulnerable to adversarial examples (Szegedy et al., 2014), i.e. maliciously modified inputs that fool a model at inference time. Many directions have been explored to explain and characterize this phenomenon (Schmidt et al., 2018; Ford et al., 2019; Ilyas et al., 2019; Shafahi et al., 2019) that became a growing concern and a major brake on the deployment of Machine Learning (ML) models. In response, many defenses have been proposed to protect the integrity of ML systems, predominantly focused on an adversary in the white-box setting (Madry et al., 2018; Zhang et al., 2019; Cohen et al., 2019; Hendrycks et al., 2019; Carmon et al., 2019). In this work, we design an innovative way to limit the transferability of adversarial perturbation towards a model, opening a new direction for robustness in the realistic black-box setting (Papernot et al., 2017). As ML-based online API are likely to become increasingly widespread, and regarding the massive deployment of edge models in a large variety of devices, several instances of a model may be deployed in systems with different environment and security properties. Thus, the black-box paradigm needs to be extensively studied to efficiently protect systems in many critical domains.

Considering a target model $M$ that a defender aims at protecting against adversarial examples, we propose a method which allows to build the model $T$, an augmented version of $M$, such that adversarial examples do not transfer from $T$ to $M$. Importantly, training $T$ only requires to have access to $M$, meaning that no labeled data set is required, so that our approach can be implemented at a low cost for any already trained model. $T$ is built by augmenting $M$ with an additional component $P$ (with $T = M \circ P$) taking the form of a neural network trained with a specific loss function with logit-based constraints. From the observation that transferability of adversarial perturbations between two models occurs because they rely on similar non-robust features (Ilyas et al., 2019), we design $P$ such that (1) the augmented network exploits useful features of $M$ and that (2) non-robust features of $T$ and $M$ are either different or require different perturbations to reach misclassification towards the same class. Our deception-based method is conceptually new as it does not aim at making $M$ relying more on robust-features as with proactive schemes (Madry et al., 2018; Zhang et al.,

2019), nor tries to anticipate perturbations which directly target the non-robust features of $M$ as with reactive strategies (Meng & Chen, 2017; Hwang et al., 2019).

Our contributions are as follows:

- We present an innovative approach to thwart transferability between two models, which we name the *luring effect*. This phenomenon, as conceptually novel, opens a new direction for adversarial research.
- We propose an implementation of the luring effect which fits any pre-trained model and does not require a label data set. An additional neural network is pasted to the target model and trained with a specific loss function that acts on the logits sequence order.
- We experimentally characterize the luring effect and discuss its potentiality for black-box defense strategies on MNIST, SVHN and CIFAR10, and analyze the scalability on ImageNet (ILSVRC2012).

For reproducibility purposes, the code is available at `https://anonymous.4open.science/r/3c64e745-927d-4f51-b187-583e64586ff6/`.

## 2 LURING ADVERSARIAL PERTURBATIONS

### 2.1 NOTATIONS

We consider a classification task where input-label pairs $(x, y) \in \mathcal{X} \times \mathcal{Y}$ are sampled from a distribution $\mathcal{D}$. $|\mathcal{Y}| = C$ is the cardinality of the labels space. A neural network model $M_\phi : \mathcal{X} \to \mathcal{Y}$, with parameters $\phi$, classifies an input $x \in \mathcal{X}$ to a label $M(x) \in \mathcal{Y}$. The pre-softmax output function of $M_\phi$ (the logits) is denoted as $h^M : \mathcal{X} \to \mathbb{R}^C$. For the sake of readability, the model $M_\phi$ is simply noted as $M$, except when necessary.

### 2.2 CONTEXT: ADVERSARIAL EXAMPLES IN THE BLACK-BOX SETTING

Black-box settings are realistic use-cases since many models are deployed (in the cloud or embedded in mobile devices) within secure environments and accessible through open or restrictive API. Contrary to the white-box paradigm where the adversary is allowed to use existing gradient-based attacks (Goodfellow et al., 2015; Carlini & Wagner, 2017; Chen et al., 2018; Dong et al., 2018; Madry et al., 2018; Wang et al., 2019), an attacker in a black-box setting only accesses the output label, confidence scores or logits from the target model. He can still take advantage of gradient-free methods (Uesato et al., 2018; Guo et al., 2019; Su et al., 2019; Brendel et al., 2018; Ilyas et al., 2018; Chen et al., 2020) but, practically, the number of queries requires to mount the attack is prohibitive and may be flagged as suspicious (Chen et al., 2019; Li et al., 2020). In that case, the adversary may take advantage of the transferability property (Papernot et al., 2017) by crafting adversarial examples on a *substitute model* and then transfering them to the target model.

### 2.3 OBJECTIVES AND DESIGN

Our objective is to find a novel way to make models more robust against transferable black-box adversarial perturbation without expensive (and sometimes prohibitive) training cost required by many white-box defense methods. Our main idea is based on classical deception-based approaches for network security (e.g. honeypots) and can be summarized as follow: *rather than try to prevent an attack, let's fool the attacker*. Our approach relies on a network $P : \mathcal{X} \to \mathcal{X}$, pasted to the already trained target network $M$ before the input layer, such as the resulting augmented model will answer $T(x) = M \circ P(x)$ when fed with input $x$. The additional component $P$ is designed and trained to reach a twofold objective:

- *Prediction neutrality:* adding $P$ does not alter the decision for a clean example $x$, i.e. $T(x) = M \circ P(x) = M(x)$;
- *Adversarial luring:* according to an adversarial example $x'$ crafted to fool $T$, $M$ does not output the same label as $T$ (i.e. $M \circ P(x') \neq M(x')$) and, in the best case, $x'$ is inefficient (i.e. $M(x') = y$).

To explain the intuition of our method, we follow the feature-based framework proposed in Ilyas et al. (2019) where a *feature* $f$ is a function from $\mathcal{X}$ to $\mathbb{R}$, that $M$ has learned to perform its predictions. Considering a binary classification task, $f$ is said to be *$\rho$-useful* ($\rho > 0$) if it satisfies Equation 1a and is *$\gamma$-useful robust* if under the worst perturbation $\delta$ chosen in a predefined set of allowed perturbations $\Delta$, $f$ stays $\gamma$-useful under this perturbation (Equation 1b). A $\rho$-useful feature $f$ is said to be a *non-robust feature* if $f$ is not robust for any $\gamma \geq 0$.

$$\text{(a)} \quad \mathbb{E}_{(x,y)\sim\mathcal{D}}\left[y \cdot f(x)\right] > \rho \quad \text{(b)} \quad \mathbb{E}_{(x,y)\sim\mathcal{D}}\left[\inf_{\delta\in\Delta} y \cdot f(x+\delta)\right] > \gamma \tag{1}$$

An adversary which aims at fooling a model will thus perform perturbations to the inputs to influence the useful features which are not robust with respect to the perturbation he is allowed to apply. We denote $\mathcal{F}_M^*$ the set of $\rho$-useful features learned by $M$. We consider a set of allowed perturbations $\Delta$ and $\gamma > 0$ such that we note $\mathcal{F}_M^{*,R}$ and $\mathcal{F}_M^{*,NR}$ respectively the set of $\gamma$-robust and non-robust features learned by $M$ (relatively to $\Delta$). An adversary that aims at fooling $M \circ P$ will alter function compositions of the form $f \circ P$ with $f \in \mathcal{F}_M^*$. These function compositions are the non-robust useful features of $M \circ P$, whose set is denoted $\mathcal{F}_{M\circ P}^{*,NR}$.

Based on the observations that transferability of adversarial perturbations between two models occurs because these models rely on similar non-robust features (Ilyas et al., 2019), we consider $f \circ P \in \mathcal{F}_{M\circ P}^{*,NR}$, and we derive two possibilities which ensure the lowest transferability between $M \circ P$ and $M$, regarding the robustness of $f$. If $f$ is robust, $f \in \mathcal{F}_M^{*,R}$, that means the adversarial perturbations from $\Delta$ is sufficient to *flip* the augmented feature $f \circ P$ but is not efficient to directly impact $f$. This case is the optimal one, since the adversarial example is unsuccessful on the target model ($M \circ P(x') \neq y$ and $M(x') = y$). On the other hand, if $f \in \mathcal{F}_M^{*,NR}$ (i.e. both $f$ and $f \circ P$ are non-robust), restraining the transferability means that the additional model $P$ impacts the way useful features vary with respect to input alterations so that the adversarial perturbation

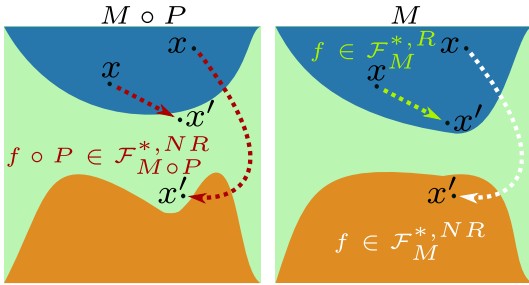

Figure 1: Luring effect. (left) $x'$ fools $M \circ P$ by flipping a non-robust feature $f \circ P$ (toward the green class). (right) However, $f$ can be a robust feature, then $M$ is not fooled (still in the blue class), or a non-robust feature but switched differently than $f \circ P$ (towards the orange class).

lead to two different labels. We encompass these two cases (illustrated in Figure 1) within what we name the luring effect. The adversary is tricked into modifying input values in some way to flip useful and non-robust features of $M \circ P$ *and* these modifications are either without effect on the useful features of $M$, or flip the non-robust features of $M$ in a different way (and therefore are detectable, as presented in Section 4).

## 2.4 TRAINING THE LURING COMPONENT

To reach our two objectives (*prediction neutrality* and *adversarial luring*), we propose to train $P$ with constraints based on the predicted labels order. For $x \in \mathcal{X}$, let $\alpha$ and $\beta$ be the labels corresponding respectively to the first and second highest confidence score given to $x$ by $M$. The training of $P$ is achieved with a new loss function that constraints $\alpha$ to (still) be the first class predicted by $M \circ P$ (*prediction neutrality*) and that makes the logits gap between $\alpha$ and $\beta$ the highest as possible for $M \circ P$ (*adversarial luring*).

To understand the intuition behind this loss function, let's formalize the concepts learned by $M$ and $M \circ P$. The prediction given by $M$ corresponds to "class $\alpha$ is predicted, class $\beta$ is the second possible class". Once $P$ has been trained following this loss function, the prediction given by $M \circ P$ corresponds to "class $\alpha$ is predicted, the higher confidence given to class $\alpha$, the smaller confidence given to class $\beta$". Concepts learned by $M$ and $M \circ P$ share the same goal of prediction, i.e. "class $\alpha$" is predicted, but the relation between class $\alpha$ and class $\beta$ is forced to be the most different as possible. As learned concepts are essentially different, then useful features learned by $M$ and $M \circ P$ to reach these concepts are necessarily different, and consequently display different types of sensitivity to

the same input pixel modifications. In other words, as the *direction* of confidence towards classes is forced to be structurally different for $M \circ P$ and $M$, we hypothesize that useful features of the two classifiers should be different and behave differently to adversarial perturbations.

The *luring loss*, designed to induce this behavior, is given in Equation 2 and the complete training procedure is detailed in Algorithm 1. The parameters of $P$ are denoted by $\theta$, $x \in \mathcal{X}$ is an input and $M$ is the target model. $M$ has already been trained and its parameters are frozen during the process. $h^M(x)$ and $h^{M \circ P}(x)$ denote respectively the logits of $M$ and $M \circ P$ for input $x$. $h_i^M(x)$ and $h_i^{M \circ P}(x)$ correspond respectively to the values of $h^M(x)$ and $h^{M \circ P}(x)$ for class $i$. The classes $a$ and $b$ correspond to the second maximum value of $h^M$ and $h^{M \circ P}$ respectively.

$$\mathcal{L}\big(x, M(x)\big) = -\lambda\big(h_{M(x)}^{M \circ P}(x) - h_a^{M \circ P}(x)\big) + max\big(0, h_b^{M \circ P}(x) - h_{M(x)}^{M \circ P}(x)\big) \qquad (2)$$

The first term of Equation 2 optimizes the gap between the logits of $M \circ P$ corresponding to the first and second biggest unscaled confidence score (logits) given by $M$ (i.e. $M(x)$ and $a$). This part formalizes the goal of changing the direction of confidence between $M \circ P$ and $M$. The second term is compulsory to reach a good classification since the first part alone does not ensure that $h_{M(x)}^{M \circ P}(x)$ is the highest logit value (*prediction neutrality*). The parameter $\lambda > 0$, called the *luring coefficient*, allows to control the trade-off between ensuring good accuracy and shifting confidence direction.

---

**Algorithm 1** Training of the luring component

---

**Input:** trained model $M$, training steps $K$, learning rate $\eta$, batch size $B$, luring coefficient $\lambda$
**Output:** luring component $P$, with parameters $\theta$
 1: $h_c^{M \circ P}(x)$ denotes the logits of $M \circ P$ for input $x$ and class $c$
 2: Randomly initialize the parameters $\theta$ of $P$
 3: **for** step $= 1 \ldots K$: **do**
 4: $\quad \{x_1, x_2, \ldots, x_B\}$: batch of training set examples
 5: $\quad$ **for** $i = 1 \ldots B$: **do**
 6: $\quad\quad (a,b) \leftarrow$ (class of the $2^{nd}$ max value of $h^M(x_i)$, class of the $2^{nd}$ max value of $h^{M \circ P}(x_i)$)
 7: $\quad\quad \mathcal{L}(x_i, M(x_i)) \leftarrow -\lambda(h_{M(x_i)}^{M \circ P}(x_i) - h_a^{M \circ P}(x_i)) + max(0, h_b^{M \circ P}(x_i) - h_{M(x_i)}^{M \circ P}(x_i))$
 8: $\quad$ **end for**
 9: $\quad \theta \leftarrow \theta - \eta \sum_{i=1}^B \nabla_\theta \mathcal{L}(x_i, M(x_i))/B$
10: **end for**
11: **return** $P$

---

## 3 CHARACTERIZATION OF THE LURING EFFECT

### 3.1 OBJECTIVE

Our first experiments are dedicated to the characterization of the luring effect by (1) evaluating our objectives in term of transferability and (2) isolating it from other factors that may influence transferability. For that purpose, we compare our approach (*Luring*) to the following approaches, which differ from ours in the training procedures and loss functions:

- *Stack* **model**: $M \circ P$ is retrained as a whole with the cross-entropy loss. *Stack* serves as a first baseline to measure the transferability between the two architectures of $M \circ P$ and $M$.

- *Auto* **model**: $P$ is an auto-encoder trained separately with binary cross-entropy loss. *Auto* serves as a second baseline of a component resulting in a *neutral* mapping from $\mathbb{R}^d$ to $\mathbb{R}^d$.

- *C_E* **model**: $P$ is trained with the cross-entropy loss between the confidence score vectors $M \circ P(x)$ and $M(x)$ in order to mimic the decision of the target model $M$. This model serves as a comparison between our loss and a loss function which does not aim at maximizing the gap between the confidence scores.

We perform experiments on MNIST (Lecun et al., 1998), SVHN (Netzer et al., 2011) and CI-FAR10 (Krizhevsky, 2009). For MNIST, $M$ has the same architecture as in Madry et al. (2018).

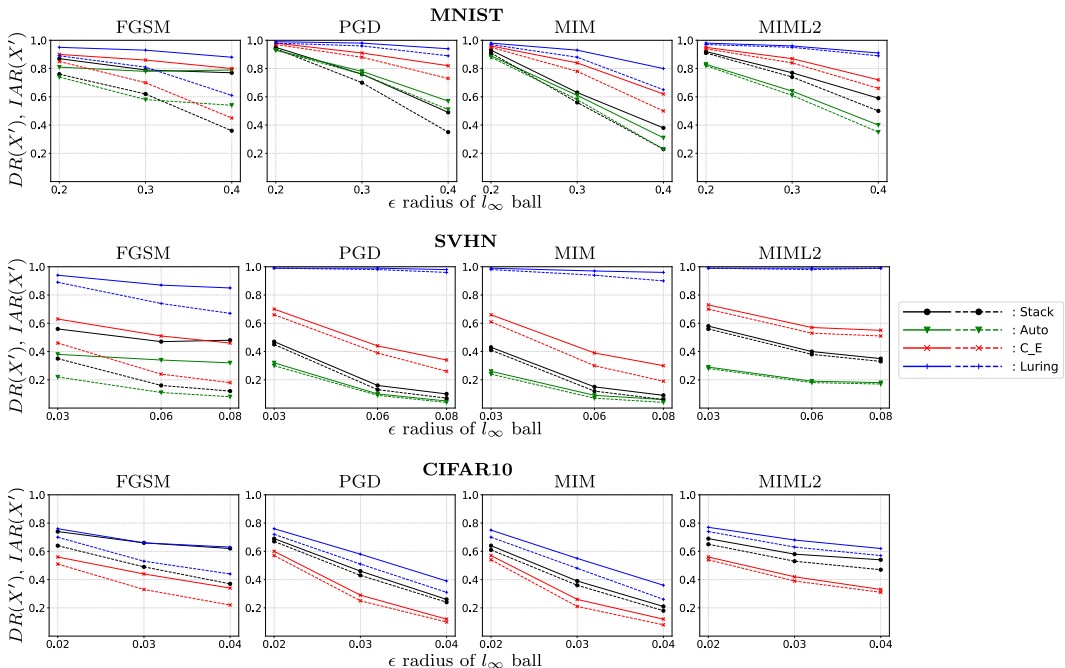

Figure 2: Disagreement Rate (solid line) and Inefficient Adversarial examples Rate (dashed line) for different attacks.

For SVHN and CIFAR10, we follow an architecture inspired from VGG (Simonyan & Zisserman, 2015). Architectures and training setup for $M$ and $P$ are detailed in Appendix A and B. Table 8 in Appendix C gathers the test set accuracy and agreement rate (on the ground-truth label) between each augmented model and $M$. We observe that our approach has a limited impact on the test accuracy with a relative decrease of 1.71%, 4.26% and 4.48% for MNIST, SVHN and CIFAR10 respectively.

## 3.2 ATTACK SETUP AND METRICS

For the characterization of the luring effect, we attack the model $M \circ P$ of the four approaches and transfer to $M$ *only* the adversarial examples that are successful for these four models. We define the *disagreement rate*, noted $DR(X')$, that represents the rate of successful adversarial examples crafted on $M \circ P$ for which $M$ and $M \circ P$ do not agree. To measure the best case where the luring effect leads to unsuccessful adversarial examples when transferred to $M$, we note $IAR(X')$ an *inefficient adversarial examples rate* that represents the proportion of successful adversarial examples on $M \circ P$ but not on $M$. For both metrics (see Equation 3), the higher, the better the luring effect to limit transferable attacks on the target model $M$.

$$DR(X') = \frac{\sum_{X'} \mathbf{1}_{M \circ P(x') \neq y, M \circ P(x') \neq M(x')}}{\sum_{X'} \mathbf{1}_{M \circ P(x') \neq y}} \quad IAR(X') = \frac{\sum_{X'} \mathbf{1}_{M \circ P(x') \neq y, M(x') = y}}{\sum_{X'} \mathbf{1}_{M \circ P(x') \neq y}} \quad (3)$$

We use the gradient-based attacks FGSM (Goodfellow et al., 2015), PGD (Madry et al., 2018), and MIM (Dong et al., 2018) in its $l_\infty$ (MIM) and $l_2$ (MIML2) versions. We used three $l_\infty$ perturbation budgets ($\epsilon$ values): 0.2, 0.3 and 0.4 for MNIST, 0.03, 0.06 and 0.08 for SVHN, and 0.02, 0.03 and 0.04 for CIFAR10. We detail the parameters used to run these attacks in Appendix D.1.

## 3.3 RESULTS AND COMPLEMENTARY ANALYSIS

We report results for the $DR$ and $IAR$ in Figure 2. The highest performances reached with our loss (for every $\epsilon$) show that our training method is efficient at inducing the luring effect. More precisely, we claim that the fact that both metrics decrease much slower as $\epsilon$ increases compared to the other

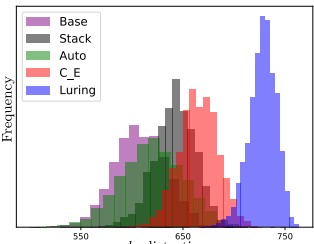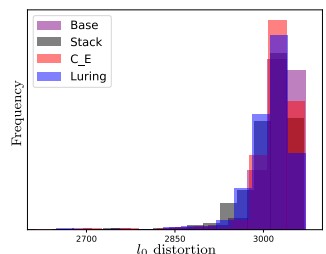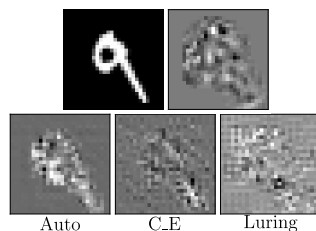

Figure 3: $l_0$ adversarial distortion for MNIST (left) and CIFAR10 (middle). Saliency maps for MNIST (right): (top) clean image and gradient of the cross-entropy loss with respect to input; (bottom) mapping gradients $\nabla_x P(x)$ for 3 augmented models.

architectures, brings additional confirmation that non-robust features of $M$ and $M \circ P$ tend to behave more differently than with the three other considered approaches.

As a complementary analysis, we investigate the magnitude of the adversarial perturbations. Results show that our approach leads to similar $l_\infty$ and $l_2$ distortion scales than the other methods. This indicates that our approach truly impacts the underlying useful features and does not only imply different distortions scales needed to fool both $M$ and $M \circ P$. The complete analysis is presented in Appendix D.2.

For MNIST, we notice that the $l_0$ distortion (i.e. the cardinality of the pixels impacted by the adversarial attack) is significantly higher with our method, as presented in Figure 3 (left). This points out that $P$ leverages more different useful features, which are easily identifiable for MNIST because of the basic structure of the images. This effect can also be demonstrated thanks to saliency maps (i.e. analyzing $\nabla_x P(x)$), as illustrated in Figure 3 (right): for the *Luring* model, $P$ is sensitive to many additional *useless* background pixels compared to $M$. Note that for *Auto*, modifying $P$'s mapping consists almost exclusively on modifying pixels correctly correlated with the true label.

For more complex inputs from SVHN or CIFAR10, without a uniform background as for MNIST, we observe that the $l_0$ distortion is approximately the same for all the architectures (Figure 3 – middle). However, for these two data sets, we analyze the influence of $P$'s mapping thanks to the logits variations. The detailed analysis in presented in Appendix D.3. We find that logits between $M$ and $M \circ P$ vary more differently with respect to input modifications when $P$ is trained with our approach, which confirms that our luring loss enables to reach Adversarial Luring.

## 4 USING THE LURING EFFECT AS A DEFENSE

By fooling an adversary on the way to target a black-box model, the phenomenon that we characterized in the previous section can be seen as a defense mechanism on its own or as a complement of state-of-the-art approaches. In this section we discuss the way to use our approach as a protection.

### 4.1 THREAT MODEL

**Attacker** Our work sets in the black-box paradigm, i.e. we assume that the adversary has no access to the inner parameters and architecture neither of the target model $M$ nor the augmented model $M \circ P$. This setting is classical when dealing with cloud-based API or edge neural networks in secure embedded systems where users have only access to input/output information with different querying abilities and output precision (i.e. scores or labels). More precisely:

- The *adversary goal* is to craft (untargeted) adversarial examples on the model he has access to, i.e. the augmented model $T = M \circ P$, and rely on transferability in order to fool $M$.

- The *adversarial knowledge* corresponds to an adversary having a black-box access to a ML system $\mathbb{S}_A$ containing the protected model $T = M \circ P$, while $M$ stays completely hidden. He can query $T$ (without any limit on the number of queries) and we assume that he is able to get the logits outputs. Moreover, for an even more strict evaluation, we also consider

      stronger adversaries allowed to use SOTA transferability designed gradient-based methods to attack $T$ (see 4.2).

- Classically, the *adversarial capability* is an upper bound $\epsilon$ of the perturbation $\|x' - x\|_\infty$.

**Defender**      For each query, the defender has access to two outputs: $M(x)$ and $M \circ P(x)$. Therefore, two different inference schemes (illustrated in Appendix E) are viable according to the goals and characteristics of the system to defend, noted $\mathbb{S}_D$:

- *(detection scenario)* If $\mathbb{S}_D$ and $\mathbb{S}_A$ are the same, the goal of the defender is to take advantage of the luring effect to detect an adversarial example by comparing the two inference outputs $M(x)$ and $M \circ P(x)$. An appropriate metric is the rate of adversarial examples which are either detected or well-predicted by $M$, as expressed in Equation 4 and noted DAC for *Detection Adversarial Accuracy*.

$$\text{DAC} = 1 - \frac{\sum_{x \in X'} \mathbf{1}_{M \circ P(x') = M(x'), M(x') \neq y}}{|X'|} \tag{4}$$

- *(transferability scenario)* If $\mathbb{S}_D$ is a secure black-box system with limited access, that only contains the target model $M$, the goal of the defender is to thwart attacks crafted from $\mathbb{S}_A$. For this specific scenario, the defender may only rely on the fact that the luring effect likely leads to unsuccessful adversarial examples ($M(x') = y$). The appropriate metric is the classical adversarial accuracy ($AC$), which is the standard accuracy measured on the adversarial set $X'$ and noted $\text{AC}_{MoP}$ and $\text{AC}_M$ respectively for $M \circ P$ and $M$.

Related real-world transferability scenarios may be linked to the increasingly widespread deployment of models in a large variety of devices (*edge AI*) or services (*cloud AI*) as an integral part of complex systems with different security requirements. For example, it is classically the case with the IoT domain where a model may be the part of a critical system and, simultaneously, be deployed and used in more mainstream connected objects with looser security requirements. The model $M$ is deployed for the critical system with strong security access, not accessible to an attacker, and the augmented protected model will be deployed in the others systems so that attacks that may be crafted more easily will not transfer to $M$.

## 4.2 ATTACKS

In order to evaluate our defense with respect to the threat model, we attack $M \circ P$ with strong gradient-free attacks:

Table 1: Adversarial accuracy for $M \circ P$ ($\text{AC}_{MoP}$), $M$ ($\text{AC}_M$), and Detection Adversarial Accuracy (DAC) for different architectures. SVHN (top), CIFAR10 (down).

| SVHN | | STACK | | | AUTO | | | C_E | | | LURING | | |
|---|---|---|---|---|---|---|---|---|---|---|---|---|---|
| | $\epsilon$ | $\text{AC}_{MoP}$ | $\text{AC}_M$ | DAC | $\text{AC}_{MoP}$ | $\text{AC}_M$ | DAC | $\text{AC}_{MoP}$ | $\text{AC}_M$ | DAC | $\text{AC}_{MoP}$ | $\text{AC}_M$ | DAC |
| SPSA | 0.03 | 0.10 | 0.54 | 0.56 | 0.06 | 0.37 | 0.38 | 0.06 | 0.67 | 0.68 | 0.0 | **0.96** | **0.97** |
| | 0.06 | 0.01 | 0.21 | 0.24 | 0.0 | 0.10 | 0.11 | 0.0 | 0.37 | 0.42 | 0.0 | **0.96** | **0.96** |
| | 0.08 | 0.0 | 0.13 | 0.15 | 0.0 | 0.06 | 0.06 | 0.0 | 0.23 | 0.28 | 0.0 | **0.94** | **0.96** |
| ECO | 0.03 | 0.06 | 0.42 | 0.44 | 0.14 | 0.48 | 0.49 | 0.18 | 0.66 | 0.68 | 0.20 | **0.97** | **0.98** |
| | 0.06 | 0.0 | 0.11 | 0.12 | 0.06 | 0.09 | 0.11 | 0.1 | 0.35 | 0.39 | 0.1 | **0.86** | **0.88** |
| | 0.08 | 0.0 | 0.03 | 0.07 | 0.06 | 0.09 | 0.09 | 0.08 | 0.29 | 0.32 | 0.09 | **0.84** | **0.86** |
| MIM-W | 0.03 | 0.04 | 0.32 | 0.35 | 0.01 | 0.20 | 0.21 | 0.03 | 0.41 | 0.45 | 0.11 | **0.81** | **0.87** |
| | 0.06 | 0.0 | 0.06 | 0.09 | 0.0 | 0.03 | 0.05 | 0.0 | 0.10 | 0.18 | 0.0 | **0.58** | **0.71** |
| | 0.08 | 0.0 | 0.03 | 0.06 | 0.0 | 0.01 | 0.02 | 0.0 | 0.06 | 0.13 | 0.0 | **0.48** | **0.67** |

| CIFAR10 | | STACK | | | C_E | | | LURING | | |
|---|---|---|---|---|---|---|---|---|---|---|
| | $\epsilon$ | $\text{AC}_{MoP}$ | $\text{AC}_M$ | DAC | $\text{AC}_{MoP}$ | $\text{AC}_M$ | DAC | $\text{AC}_{MoP}$ | $\text{AC}_M$ | DAC |
| SPSA | 0.02 | 0.01 | 0.75 | 0.78 | 0.06 | 0.68 | 0.71 | 0.12 | **0.78** | **0.82** |
| | 0.03 | 0.0 | 0.57 | 0.62 | 0.01 | 0.45 | 0.49 | 0.02 | **0.59** | **0.64** |
| | 0.04 | 0.0 | 0.38 | 0.42 | 0.0 | 0.22 | 0.24 | 0.01 | **0.42** | **0.50** |
| ECO | 0.02 | 0.15 | 0.7 | 0.71 | 0.14 | 0.6 | 0.63 | 0.16 | **0.81** | **0.87** |
| | 0.03 | 0.09 | 0.44 | 0.59 | 0.09 | 0.36 | 0.42 | 0.2 | **0.65** | **0.67** |
| | 0.04 | 0.05 | 0.29 | 0.29 | 0.04 | 0.28 | 0.34 | 0.09 | **0.35** | **0.44** |
| MIM-W | 0.02 | 0.0 | 0.49 | 0.52 | 0.02 | 0.4 | 0.43 | 0.03 | **0.58** | **0.64** |
| | 0.03 | 0.0 | 0.24 | 0.28 | 0.0 | 0.15 | 0.19 | 0.0 | **0.30** | **0.39** |
| | 0.04 | 0.0 | 0.13 | 0.16 | 0.0 | 0.05 | 0.1 | 0.0 | **0.18** | **0.28** |

- **SPSA** attack (Uesato et al., 2018) ensures to consider the strongest adversary in our black-box setting as it allows the adversary to get the logits of $M \circ P$.

- Coherently with an adversary that has no querying limitation, **ECO** (Moon et al., 2019) is a strong score-based gradient-estimation free attack.

To perform an even more strict evaluation, and to anticipate future gradient-free attacks, we report the best results[1] obtained with the state-of-the-art transferability designed gradient-based attacks **MIM**, **DIM**, **MIM-TI** and **DIM-ti** (Dong et al., 2018; Xie et al., 2018; Dong et al., 2019), under the name **MIM-W**. The parameters used to run these attacks are presented in Appendix F.

### 4.3    RESULTS

The results are presented in Table 1 for SVHN and CIFAR10 (results for MNIST are close to SVHN and are presented in Appendix G). For CIFAR10, since no autoencoder we tried allows to reach a correct test set accuracy for the *Auto* approach, we do not consider it. First, we notice that $AC_{MOP}$ almost always equal 0 with gradient-based iterative attacks[2]. This is an indication that $P$ does not induce a form of gradient masking (Athalye et al., 2018).
Remarkably on SVHN, for $\epsilon = 0.08$ (the largest perturbation), and for the worst-case attacks, adversarial examples tuned to achieve the best transferability only reduce $AC_M$ to 0.48 against our approach, compared to almost 0 for the other architectures. The robustness benefits are more observable on SVHN and MNIST but the results on CIFAR10 are particularly promising in the scope of a defense scheme that only requires a pre-trained model. Indeed, for the common $l_\infty$ perturbation value of 0.03, the worst DAC value for the *C_E* and *Luring* approaches are respectively 0.19 (against DIM attack) and 0.39 (against DIM-TI attack).

### 4.4    COMPATIBILITY WITH IMAGENET AND ADVERSARIAL TRAINING

We scale our approach to ImageNet (ILSVRC2012). For our experiments, the target model $M$ consists of a MobileNetV2 model (Sandler et al., 2018), reaching 71.3% of accuracy on the validation set. For a common $l_\infty$ perturbation budget of 4/255, the smaller $AC_M$ and $DAC$ observed against the strong **MIM-W** attack with our approach equal 0.4 and 0.55, while they equal 0.23 and 0.35 with the *C_E* approach. Following the results previously observed on the benchmarks used for characterization, these results validate the scalability of our approach to large-scale data sets. Details on the ImageNet experiments and results are presented in Appendix H.1

Even if very few efforts tackles the issue of thwarting adversarial perturbations transferred from a source model to a target model (transfer black-box attacks), there exists numerous detection and defense schemes intended to protect a target model in a white-box or gray-box setting against adversarial perturbations. As the *luring effect* is conceptually new, it can be combined to any of these existing methods, to provide even more protection. Indeed, our deceiving-based approach acts on the way $M \circ P$ performs inference relatively to $M$. As the effort is focused on the design of $P$, another defense method, this time focusing on $M$ and designed to protect $M$, can then be combined with our scheme. We choose here to consider the combination with adversarial training (Madry et al., 2018), a state-of-the-art approach for robustness in the white-box setting. The model $M$ is already trained with adversarial training. Interestingly, we note that the joint use of these defenses clearly improves the detection performance, with DAC values superior to 0.8 for the three data sets as well a strong improvement of the $AC_M$ metric for MNIST (0.97) and CIFAR10 (0.85). The detailed experiments are presented in Appendix H.2.

## 5    DISCUSSION AND RELATED WORK

A good practice in the field of adversarial robustness is to consider an adaptive adversary (Carlini et al., 2019): an attacker using some knowledge of the defense method to bypass it more efficiently. In our case, the only information that could be given to an adversary without breaking the black-box

---

[1]Here *best* is from the adversary point of view.

[2]Parameters are tuned relatively to the adversarial goal: lowering our defense efficiency. Thus, the lowest performances sometimes do not correspond to the lowest accuracy of $M \circ P$ on the adversarial examples.

threat model is the supplementary knowledge that the augmented model has been trained with the luring loss. Indeed, M has to stay hidden, otherwise the adversary could use a decision-based attack to thwart the defense (Brendel et al., 2018; Cheng et al., 2020). Moreover, with a more permissive access to $T$, for example a white-box access to $T$, the adversary could simply extract $M$ from the augmented model, and then defeat the defense. The additional information of the luring loss, would not bring any advantage to the adversary in order to craft adversarial perturbations on $T$ which transfer to $M$. Indeed, it would imply for the attacker to inverse the effect of the luring loss on the prediction output vector of $T$, which appears as a prohibitive effort. Therefore, respecting the threat model, an adaptive adversary would not be more efficient than the adversary considered throughout this work.

Our characterization and first results pave the way for a possible extension of the luring effect to more permissive threat models. For now, in a white-box setting, the defense scheme would be defeated since the adversary could use a gradient-based attack to fool both $M$ and $M \circ P$. Actually, the better solution to exploit the luring effect in a gray-box setting would be to cause the luring effect between two models $M$ and $M'$ with different architectures. A model $M'$ trained with the luring loss would be publicly released. The adversary could then be allowed to mount gradient-based attacks on $M'$ without being able to access $M$. The possible efficiency of such future work is backed by results on the MIM-W attacks, which corresponds to $T$ being attacked by gradient-based attacks.

Defenses in the black-box context are weakly covered in the literature as compared to the numerous approaches focused on white-box attacks. More particularly, very few approaches deal with query-based black-box attacks, such as the recent BlackLight (Li et al., 2020). To our knowledge, our work is the first to exploit the idea of luring an adversary taking advantage of the transferability property. However, a honeypot-based approach has been recently proposed (Shan et al., 2019) with a *trapdoored model* to detect adversarial examples. Even if the threat model (white-box) is different from ours, this approach is conceptually close to our deception approach, and it is interestingly mentioned that transferability between the target and the trapdoor-protected model is very poor. By fitting the trapdoor approach in our black-box threat model we highlight complementary performances in terms of $AC_M$ and $DAC$ metrics, meaning that an overall deception-based strategy is a promising direction for future work. The detailed analysis is presented in Appendix I.

## 6 CONCLUSION

We propose a conceptually innovative approach to improve the robustness of a model against transfer black-box adversarial perturbations, which basically relies on a deception strategy. Inspired by the notion of robust and non-robust features, we derive and characterize the *luring effect*, which is implemented via a decoy network built upon the target model, and a loss designed to fool the adversary into targeting different non-robust features than the ones of the target model. Importantly, this approach only relies on the logits of target model, does not require a labeled data set and therefore can by applied to any pre-trained model.

We show that our approach can be used as a defense and that a defender may exploit two prediction schemes to detect adversarial examples or enhance the adversarial robustness. Experiments on MNIST, SVHN, CIFAR10 and ImageNet demonstrate that exploiting the luring effect enables to successfully thwart an adversary using state-of-the-art optimized attacks even with large adversarial perturbations.

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

## A  Setup for base classifiers

The architectures of the models trained on MNIST, SVHN and CIFAR10 are detailed respectively in tables 2, 3 and 4. BN, MaxPool(u,v), UpSampling(u,v) and Conv(f,k,k) denote respectively batch normalization, max pooling with window size (u,v), upsampling with sampling factor (u,v) and 2D convolution with f filters and kernel of size (k,k).

For MNIST, we used 5 epochs, a batch size of 28 and the Adam optimizer with a learning rate of 0.01. For SVHN, we used 50 epochs, a batch size of 28 and the Adam optimizer with a learning rate of 0.01. For CIFAR10, we used 200 epochs, a batch size of 32, and the Adam optimizer with a piecewise learning rate of 0.1, 0.01 and 0.001 after respectively 80 and 120 epochs.

Table 2: MNIST base classifier architecture. Epochs: 5. Batch size: 28. Optimizer: Adam with learning rate 0.01.

| ARCHITECTURE |
| --- |
| CONV(32,5,5) + RELU |
| MAXPOOL(2,2) |
| CONV(64,5,5) + RELU |
| MAXPOOL(2,2) |
| DENSE(1024) + RELU |
| DENSE(10) + SOFTMAX |

Table 3: SVHN base classifier architecture. Epochs: 5. Batch size: 28. Optimizer: Adam with learning rate 0.01.

| ARCHITECTURE |
| --- |
| CONV(64,3,3) + BN + RELU |
| CONV(64,3,3) + MAXPOOL(2,2) + BN + RELU |
| CONV(128,3,3) + BN + RELU |
| CONV(128,3,3) + MAXPOOL(2,2) + BN + RELU |
| CONV(256,3,3) + BN + RELU |
| CONV(256,3,3) + MAXPOOL(2,2) + BN + RELU |
| DENSE(1024) + BN + RELU |
| DENSE(1024) + BN + RELU |
| DENSE(10) + SOFTMAX |

Table 4: CIFAR10 base classifier architecture. Epochs: 200. Batch size: 32. Optimizer: Adam with learning rate starting at 0.1, decreasing to 0.01 and 0.001 respectively after 80 and 120 epochs.

| ARCHITECTURE |
| --- |
| CONV(128,3,3) + BN + RELU |
| CONV(128,3,3) + MAXPOOL(2,2) + BN + RELU |
| CONV(256,3,3) + BN + RELU |
| CONV(256,3,3) + MAXPOOL(2,2) + BN + RELU |
| CONV(512,3,3) + BN + RELU |
| CONV(512,3,3) + MAXPOOL(2,2) + BN + RELU |
| DENSE(1024) + BN + RELU |
| DENSE(1024) + BN + RELU |
| DENSE(10) + SOFTMAX |

## B  Training setup for defense components

The architectures for the defense component $P$ on MNIST, SVHN and CIFAR10 are detailed respectively in Tables 5, 6 and 7. $BN$, $MaxPool(u,v)$, $UpSampling(u,v)$ and $Conv(f,k,k)$ denote respectively batch normalization, max pooling with window size $(u,v)$, upsampling with sampling factor $(u,v)$ and 2D convolution with $f$ filters and kernel of size $(k,k)$. The detailed parameters used to perform training are also reported.

Table 5: Defense component architecture for MNIST.

| ARCHITECTURE |
| --- |
| CONV(16,3,3) + RELU |
| MAXPOOL(2,2) |
| CONV(8,3,3) + RELU |
| MAXPOOL(2,2) |
| CONV(8,3,3) + RELU |
| CONV(8,3,3) + RELU |
| UPSAMPLING(2,2) |
| CONV(8,3,3) + RELU |
| UPSAMPLING(2,2) |
| CONV(16,3,3) |
| UPSAMPLING(2,2) |
| CONV(1,3,3) + SIGMOID |

**MNIST:**

- *Stack* Epochs: 5. Batch size: 28. Optimizer: Adam with learning rate 0.001

- *Auto* Epochs: 50. Batch size: 128. Optimizer: Adam with learning rate 0.001

- *C_E* Epochs: 64. Batch size: 64. Optimizer: Adam with learning rate starting at 0.001, decreasing to 0.0002 and 0.0004 respectively after 45 and 58 epochs

- *Luring* Epochs: 64. Batch size: 64. Optimizer: Adam with learning rate starting at 0.001, decreasing to 0.0002 and 0.0004 respectively after 45 and 58 epochs

Table 6: Defense component architecture for SVHN.

| ARCHITECTURE |
| --- |
| CONV(128,3,3) + BN + RELU |
| MAXPOOL(2,2) |
| CONV(256,3,3) + BN + RELU |
| MAXPOOL(2,2) |
| CONV(512,3,3) + BN + RELU |
| CONV(1024,3,3) + BN + RELU |
| MAXPOOL(2,2) |
| CONV(512,3,3) + BN + RELU |
| CONV(512,3,3) + BN + RELU |
| UPSAMPLING(2,2) |
| CONV(256,3,3) + BN + RELU |
| UPSAMPLING(2,2) |
| CONV(128,3,3) + BN + RELU |
| UPSAMPLING(2,2) |
| ONV(64,3,3) + BN + RELU |
| CONV(3,3,3) + BN + SIGMOID |

**SVHN:**

- *Stack* Epochs: 20. Batch size: 256. Optimizer: Adam with learning rate 0.001

- *Auto* Epochs: 5. Batch size: 128. Optimizer: Adam with learning rate 0.001

- *C_E* Epochs: 210. Batch size: 256. Optimizer: Adam with learning rate starting at 0.0001, decreasing to 0.00001 and 0.000008 respectively after 126 and 168 epochs

- *Luring* Epochs: 210. Batch size: 256. Optimizer: Adam with learning rate starting at 0.0001, decreasing to 0.00001 and 0.000008 respectively after 126 and 168 epochs. Dropout is used

Table 7: Defense component architecture for CIFAR10.

| ARCHITECTURE |
| --- |
| CONV(128,3,3) + BN + RELU |
| CONV(256,3,3) + BN + RELU |
| MAXPOOL(2,2) |
| CONV(512,3,3) + BN + RELU |
| CONV(1024,3,3) + BN + RELU |
| CONV(512,3,3) + BN + RELU |
| CONV(512,3,3) + BN + RELU |
| CONV(256,3,3) + BN + RELU |
| UPSAMPLING(2,2) |
| CONV(128,3,3) + BN + RELU |
| CONV(64,3,3) + BN + RELU |
| CONV(3,3,3) + BN + SIGMOID |

**CIFAR10:**

- *Stack* Epochs: 200. Batch size: 32. Optimizer: Adam with learning rate starting at 0.1, decreasing to 0.01 and 0.001 respectively after 80 and 120 epochs
- *C_E* Epochs: 216. Batch size: 256. Optimizer: Adam with learning rate starting at 0.00001, decreasing to 0.000005 and 0.0000008 respectively after 154 and 185 epochs. Dropout is used
- *Luring* Epochs: 216. Batch size: 256. Optimizer: Adam with learning rate starting at 0.00001, decreasing to 0.000005 and 0.0000008 respectively after 154 and 185 epochs. Dropout is used

## C  TEST SET ACCURACY AND AGREEMENT RATES

For the luring parameter, we set $\lambda = 1$ for MNIST and SVHN, and $\lambda = 0.15$ for CIFAR10. For CIFAR10, since no autoencoder we tried reaches correct test set accuracy, we exclude the *Auto* model.

Table 8: Test set accuracy and agreement (augmented and target model agree on the ground-truth label) between each augmented model and the target model $M$, noted BASE.

| MODEL | DATA SET | | | | | |
| --- | --- | --- | --- | --- | --- | --- |
| | MNIST | | SVHN | | CIFAR10 | |
| | TEST | AGREE | TEST | AGREE | TEST | AGREE |
| BASE | 0.991 | – | 0.961 | – | 0.893 | – |
| STACK | 0.98 | 0.976 | 0.925 | 0.913 | 0.902 | 0.842 |
| AUTO | 0.971 | 0.969 | 0.95 | 0.943 | – | – |
| C_E | 0.982 | 0.977 | 0.919 | 0.907 | 0.860 | 0.834 |
| LURING | 0.974 | 0.969 | 0.920 | 0.917 | 0.853 | 0.822 |

## D  LURING EFFECT ANALYSIS

### D.1  ATTACK PARAMETERS

For MIML2, we report results when adversarial examples are clipped to respect the threat model with regards to $\epsilon$. An illustration of a clean image and its adversarial counterpart for the maximum perturbation allowed is presented in Figure 4. We note that the ground-truth label is still clearly recognizable. For PGD, MIM and MIML2, the number of iterations is set to 1000, the step size to 0.01 and $\mu$ to 1.0 (MIM and MIML2). For MIML2, the $l_2$ bound is set to 30 on MNIST and 2 on SVHN and CIFAR10.

### D.2  ANALYSIS OF THE $l_2$ AND $l_\infty$ DISTORTIONS

We investigate the impact of our additional component on the distance between a clean and an adversarial example, compared to the other approaches. We measure the $l_2$ means of the adversarial

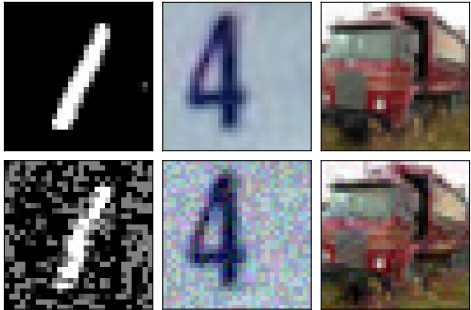

Figure 4: (top) Clean image, (bottom) adversarial example for the maximum perturbation allowed (left to right: $\epsilon = 0.4, 0.08, 0.04$).

perturbations needed to find an adversarial example, and verify that our approach is globally at the same level than the other approaches.

For that purpose, an appropriate and recommended attack is the Carlini & Wagner $l_2$ attack (hereafter, CWL2) (Carlini & Wagner, 2017) that searches for the closest adversarial examples in terms of the $l_2$ distortion. We run CWL2 with the following parameters: 10 binary search steps, learning rate of 0.1, initial constant of 0.5 and 500 iterations for MNIST, SVHN and CIFAR10. These parameters have been chosen such that increasing the number of iterations does not lower the final $l_2$ distortion. For each data set, we run the attack on $1,000$ test set examples correctly classified for the four approaches for both $M$ and $M \circ P$.

Table 9: Average $l_2$ and $l_\infty$ distortions between clean and adversarial examples generated with the CWL2 attack for the target model $M$ and the augmented models $M \circ P$

| MODEL | MNIST | | SVHN | | CIFAR10 | |
|---|---|---|---|---|---|---|
| | MEAN $l_2$ | MEAN $l_\infty$ | MEAN $l_2$ | MEAN $l_\infty$ | MEAN $l_2$ | MEAN $l_\infty$ |
| TARGET | 0.96 | 0.27 | 0.32 | 0.04 | 0.42 | 0.04 |
| STACK | 0.87 | 0.25 | 0.34 | 0.05 | 0.4 | 0.03 |
| AUTO | 0.94 | 0.35 | 0.29 | 0.04 | – | – |
| C_E | 0.92 | 0.34 | 0.29 | 0.04 | 0.41 | 0.04 |
| LURING | 0.94 | 0.31 | 0.31 | 0.04 | 0.39 | 0.04 |

The average $l_2$ and $l_\infty$ distortions are reported in Table 9 and show that there is no significant difference between our approach and the other approaches. This is an additional observation that our *luring loss* allows to train an augmented model which causes the *luring effect* by predominantly targeting different useful non-robust features rather than artificially modifying the scale of the adversarial distortion needed to cause misclassification.

### D.3 LOGITS ANALYSIS

As a complementary analysis on the impact of the mapping induced by the additional compoment $P$, we analyse how $P$ acts on the variation of the logits with respect to the input features ($\nabla_{x_i} \ell(x)$). For that purpose, we first pick 1000 test set examples correctly classified by all the models (the base model $M$ and the augmented models of the four approaches: *Auto*, *Stack*, *C_E* and *Luring*). Then, for each augmented model and each class $c$, we compute the number $\Omega_c$ of pixels $x_d$ that lead to opposite effect on prediction between $M$ and $M \circ P$ (see Equation 5):

$$\Omega_c = \left| \left\{ d : \frac{\partial \ell_c}{\partial x_d} \cdot \frac{\partial \ell\, P_c}{\partial x_d} < 0 \right\} \right| \tag{5}$$

Next, we look at the proportion of these examples for which $\Omega_c$ is higher for the Luring approach than for the other approaches. Results for each class are presented in Table 10. For both SVHN and CIFAR10, these proportions are strictly higher than $80\%$ whatever the class. This is a supplementary

indication toward the fact that – with respect to the allowed adversarial perturbation altering $x$ – our approach is more relevant than other approaches (that also perform a mapping within $\mathcal{X}$) to *flip* useful and non-robust features of $M \circ P$ and $M$ towards different labels.

Table 10: Rate of examples (over 1000) for which logits vary more differently between $M$ and $M \circ P$, for the *Luring* approach against the other approaches.

| | CLASS | | | | | | | | | |
|---|---|---|---|---|---|---|---|---|---|---|
| | 1 | 2 | 3 | 4 | 5 | 6 | 7 | 8 | 9 | 10 |
| SVHN | 0.81 | 0.85 | 0.86 | 0.89 | 0.86 | 0.82 | 0.83 | 0.84 | 0.83 | 0.82 |
| CIFAR | 0.86 | 0.88 | 0.88 | 0.87 | 0.88 | 0.87 | 0.89 | 0.86 | 0.88 | 0.85 |

## E   THREAT MODELS ILLUSTRATION

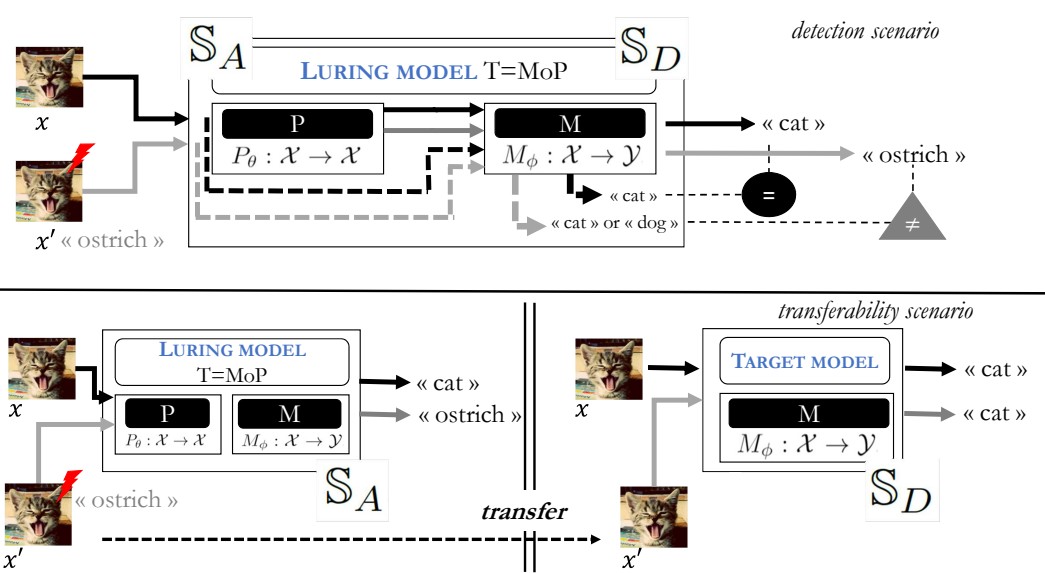

Figure 5: Illustration of the black-box threat model with two scenarios. (top) In the *inner transferability scenario* the system to attack and to defend is the same. Because the luring effect induces different behavior when facing adversarial perturbation, the defender is able to detect adversarial examples by comparing $M(x)$ and $T(x)$. (bottom) In the *distant transferability scenario*, the system to defend may suffer from transferable adversarial examples but the defender takes advantage of the weak transferability between $T$ and $M$.

## F   ATTACK PARAMETERS

All the attacks (SPSA, ECO, MIM, DIM, MIM-TI, DIM-TI) are performed on 1000 correctly classified test set examples, with parameter values tuned for the highest transferability results. More precisely, we searched for parameters leading to the lowest $\mathrm{AC}_M$ and DAC values.

## F.1 GRADIENT-FREE ATTACKS

For the SPSA attack, the learning rate is set to $0.1$, the number of iterations is set to $100$ and the batch size to $128$. For the ECO attack, for the three data sets, the number of queries is set to $20,000$. We did not perform early-stopping as it results in less transferable adversarial examples. The block size is set to $2$, $8$ and $4$ respectively for MNIST, SVHN and CIFAR10.

## F.2 GRADIENT-BASED ATTACKS

For DIM and DIM-TI, the optimal $p$ value was searched in $\{0.1, 0.2, \cdots, 1.0\}$. We obtained the optimal values of $1.0$, $1.0$ and $0.6$ respectively on MNIST, SVHN and CIFAR10. For MIM and its variants, the number of iterations is set to $1000$, $500$ and $100$ respectively on MNIST, SVHN and CIFAR10, and $\mu = 1.0$. The optimal kernel size required by MIM-TI and DIM-TI was searched in $\{(3, 3), (5, 5), (10, 10), (15, 15)\}$. For the MIM-TI and DIM-TI, the size of the kernel resulting in the lowest $AC_M$ and DAC values reported in Section 4 are presented in tables 11, 12 and 13 respectively for MNIST, SVHN and CIFAR10.

Table 11: MNIST. Kernel size for the MIM-TI and DIM-TI attacks.

| ATTACK | ARCHITECTURE | $\epsilon$ VALUE | | |
|---|---|---|---|---|
| | | 0.3 | 0.4 | 0.5 |
| MIM-TI | STACK | $5 \times 5$ | $5 \times 5$ | $5 \times 5$ |
| | AUTO | $10 \times 10$ | $5 \times 5$ | $5 \times 5$ |
| | C_E | $10 \times 10$ | $10 \times 10$ | $5 \times 5$ |
| | LURING | $5 \times 5$ | $5 \times 5$ | $10 \times 10$ |
| DIM-TI | STACK | $5 \times 5$ | $5 \times 5$ | $5 \times 5$ |
| | AUTO | $5 \times 5$ | $5 \times 5$ | $5 \times 5$ |
| | C_E | $10 \times 10$ | $10 \times 10$ | $10 \times 10$ |
| | LURING | $5 \times 5$ | $5 \times 5$ | $5 \times 5$ |

Table 12: SVHN. Kernel size for the MIM-TI and DIM-TI attacks.

| ATTACK | ARCHITECTURE | $\epsilon$ VALUE | | |
|---|---|---|---|---|
| | | 0.03 | 0.06 | 0.08 |
| MIM-TI | STACK | $5 \times 5$ | $5 \times 5$ | $5 \times 5$ |
| | AUTO | $5 \times 5$ | $5 \times 5$ | $5 \times 5$ |
| | C_E | $5 \times 5$ | $10 \times 10$ | $5 \times 5$ |
| | LURING | $10 \times 10$ | $10 \times 10$ | $10 \times 10$ |
| DIM-TI | STACK | $5 \times 5$ | $5 \times 5$ | $5 \times 5$ |
| | AUTO | $5 \times 5$ | $5 \times 5$ | $5 \times 5$ |
| | C_E | $5 \times 5$ | $5 \times 5$ | $5 \times 5$ |
| | LURING | $10 \times 10$ | $10 \times 10$ | $10 \times 10$ |

Table 13: CIFAR10. Kernel size for the MIM-TI and DIM-TI attacks.

| ATTACK | ARCHITECTURE | $\epsilon$ VALUE | | |
|---|---|---|---|---|
| | | 0.02 | 0.03 | 0.04 |
| MIM-TI | STACK | $3 \times 3$ | $3 \times 3$ | $3 \times 3$ |
| | AUTO | $3 \times 3$ | $3 \times 3$ | $3 \times 3$ |
| | C_E | $3 \times 3$ | $10 \times 10$ | $3 \times 3$ |
| | LURING | $3 \times 3$ | $3 \times 3$ | $3 \times 3$ |
| DIM-TI | STACK | $3 \times 3$ | $3 \times 3$ | $3 \times 3$ |
| | AUTO | $3 \times 3$ | $3 \times 3$ | $3 \times 3$ |
| | C_E | $3 \times 3$ | $3 \times 3$ | $3 \times 3$ |
| | LURING | $3 \times 3$ | $3 \times 3$ | $3 \times 3$ |

## G    RESULT FOR MNIST

Table 14: MNIST. $AC_{MoP}$, $AC_M$ and DAC for different source model architectures.

|  |  | STACK | | | AUTO | | | C_E | | | LURING | | |
|---|---|---|---|---|---|---|---|---|---|---|---|---|---|
|  | $\epsilon$ | $AC_{MoP}$ | $AC_M$ | DAC | $AC_{MoP}$ | $AC_M$ | DAC | $AC_{MoP}$ | $AC_M$ | DAC | $AC_{MoP}$ | $AC_M$ | DAC |
| SPSA | 0.2 | 0.0 | 0.96 | 0.97 | 0.03 | 0.95 | 0.95 | 0.0 | 0.97 | 0.98 | 0.14 | **0.99** | **0.99** |
|  | 0.3 | 0.0 | 0.86 | 0.89 | 0.03 | 0.90 | 0.92 | 0.0 | 0.94 | 0.95 | 0.05 | **0.96** | **0.97** |
|  | 0.4 | 0.0 | 0.72 | 0.77 | 0.0 | 0.85 | 0.87 | 0.0 | 0.88 | 0.91 | 0.02 | **0.95** | **0.96** |
| ECO | 0.2 | 0.03 | 0.86 | 0.92 | 0.03 | 0.86 | 0.88 | 0.01 | 0.91 | 0.93 | 0.05 | **0.99** | **1.0** |
|  | 0.3 | 0.02 | 0.56 | 0.65 | 0.03 | 0.68 | 0.7 | 0.01 | 0.8 | 0.87 | 0.02 | **0.91** | **0.96** |
|  | 0.4 | 0.01 | 0.35 | 0.54 | 0.03 | 0.36 | 0.46 | 0.01 | 0.45 | 0.48 | 0.03 | **0.77** | **0.79** |
| MIM-W | 0.2 | 0.0 | 0.79 | 0.82 | 0.0 | 0.81 | 0.82 | 0.0 | 0.85 | 0.88 | 0.19 | **0.92** | **0.93** |
|  | 0.3 | 0.0 | 0.31 | 0.45 | 0.0 | 0.35 | 0.45 | 0.0 | 0.43 | 0.57 | 0.13 | **0.69** | **0.75** |
|  | 0.4 | 0.0 | 0.07 | 0.24 | 0.0 | 0.07 | 0.17 | 0.0 | 0.13 | 0.31 | 0.07 | **0.34** | **0.45** |

## H    IMAGENET AND ADVERSARIAL TRAINING

### H.1    RESULTS ON IMAGENET

The ImageNet data set (Deng et al., 2009) is composed of color images, with $1.2$ million images and $50,000$ samples in the training and validation set respectively. For our experiments, the target model $M$ consists of a MobileNetV2 model (Sandler et al., 2018), taking inputs of size $224 \times 224 \times 3$ scales between $-1$ and $1$. The architecture for the defense component is described in Table 15. We choose to present our approach along with the $C\_E$ approach as it represents well the purpose of the method we present in this paper. Indeed, the $C\_E$ approach only requires training the additional component $P$, which fits into the scope of a defense which can be implemented on top of any already trained model $M$.

Table 15: Defense component architecture for ImageNet.

| ARCHITECTURE |
|---|
| CONV(128,3,3) + BN + RELU |
| MAXPOOL(2,2) |
| CONV(256,3,3) + BN + RELU |
| MAXPOOL(2,2) |
| CONV(512,3,3) + BN + RELU |
| MAXPOOL(2,2) |
| CONV(512,3,3) + BN + RELU |
| CONV(256,3,3) + BN + RELU |
| UPSAMPLING(2,2) |
| CONV(128,3,3) + BN + RELU |
| UPSAMPLING(2,2) |
| CONV(64,3,3) + BN + RELU |
| CONV(3,3,3) + BN + SIGMOID |

For both the *Luring* and $C\_E$ approaches, the additional component $P$ is trained for $100$ epochs, with a bath size of $256$, using a momentum of $0.9$ with learning rate starting at $0.01$, decreasing to $0.001$ after $80$ epochs. The $\lambda$ parameter of the loss is set to $1.0$. The accuracy of the augmented model as well as the agreement rate (the model $M$ and the augmented model $M \circ P$) are presented in Table 16.

Table 16: Test set accuracy and agreement (augmented and target model agree on the ground-truth label) between each augmented model and the target model $M$, noted BASE.

| MODEL | TEST | AGREE |
|---|---|---|
| BASE | 0.713 | – |
| C_E | 0.679 | 0.647 |
| LURING | 0.653 | 0.614 |

The attacks are performed on $1,000$ well-classified images from the validation set, and two common $l_\infty$ perturbation budgets are considered : $4/255$, $5/255$ and $6/255$. Results are presented in Table 17. The highest results observed in terms of both $AC_M$ and $DAC$ values with our approach confirm the scalability of our method on large-scale data sets.

Table 17: ImageNet. $AC_{MoP}$, $AC_M$ and DAC for different source model architectures.

|  | $\epsilon$ | C_E | | | LURING | | |
|---|---|---|---|---|---|---|---|
|  |  | $AC_{MoP}$ | $AC_M$ | DAC | $AC_{MoP}$ | $AC_M$ | DAC |
| MIM-W | 4/255 | 0.0 | 0.23 | 0.35 | 0.00 | **0.4** | **0.55** |
|  | 5/255 | 0.0 | 0.15 | 0.25 | 0.00 | **0.28** | **0.43** |
|  | 6/255 | 0.0 | 0.08 | 0.18 | 0.00 | **0.18** | **0.33** |

## H.2 COMPATIBILITY WITH ADVERSARIAL TRAINING

We present the performances of the *luring* approach when the target model $M$ is trained with adversarial training (Madry et al., 2018) as a general state-of-the-art method for adversarial robustness. The $l_\infty$ bound constraint for PGD is set to $0.3$ for MNIST and $0.03$ for both SVHN and CIFAR10. During training, the PGD is performed for $40$ steps, and a step size of $0.1$ for MNIST. The PGD is performed for $10$ steps with a step size of $0.008$ for both SVHN and CIFAR10. We report in Table 18 the accuracy and agreement (augmented and target models agree on the ground-truth label) between the target model $M$ and its augmented version $T$, whether $M$ is trained classically or with adversarial training.

In Table 19 we report the $AC_M$ and $DAC$ values corresponding to the same threat model as the one used throughout this paper, against the strong **MIM-W** attack. Moroever, we add the accuracy in a white-box setting of the model $M$ against the PGD attack with $40$ iterations and a step size of $0.01$, denoted as $AC_{M,wb}$. Interestingly, we observe a productive interaction between the two defenses with $DAC$ values that have greatly increased for the three data sets, as well as the $AC_M$ values for MNIST and CIFAR10.

Table 18: Test set accuracy and agreement (augmented and target model agree on the ground-truth label) between an augmented luring model $T$ and a target model $M$. Base and Base–Luring denote respectively a target model trained classically and the augmented model trained with the luring defense. AdvTrain and AdvTrain–Luring denote respectively a target model trained with adversarial training and the augmented model trained with the luring defense.

| MODEL | DATA SET | | | | | |
|---|---|---|---|---|---|---|
|  | MNIST | | SVHN | | CIFAR10 | |
|  | TEST | AGREE | TEST | AGREE | TEST | AGREE |
| BASE | 0.991 | – | 0.961 | – | 0.893 | – |
| BASE–LURING | 0.97 | 0.97 | 0.92 | 0.92 | 0.85 | 0.82 |
| ADVTRAIN | 0.99 | – | 0.93 | – | 0.79 | – |
| ADVTRAIN–LURING | 0.96 | 0.96 | 0.87 | 0.85 | 0.78 | 0.76 |

Table 19: $AC_{M,wb}$, $AC_M$ and DAC values (among FGSM, MIM, DIM, MIM-TI and DIM-TI) for the luring approach. Base–Luring and AdvTrain–luring denote the cases where the model $M$ is trained respectively classically and with adversarial training.

|  | $\epsilon$ | LURING | | | ADVTRAIN–LURING | | |
|---|---|---|---|---|---|---|---|
|  |  | $AC_{M,wb}$ | $AC_M$ | DAC | $AC_{M,wb}$ | $AC_M$ | DAC |
| MNIST | 0.3 | 0.0 | 0.69 | 0.75 | 0.82 | 0.97 | 0.98 |
| SVHN | 0.03 | 0.0 | 0.81 | 0.87 | 0.45 | 0.898 | 0.911 |
| CIFAR10 | 0.03 | 0.0 | 0.30 | 0.39 | 0.41 | 0.85 | 0.88 |

# I RELATED WORK: COMBINING WITH TRAPDOORED MODELS

In Shan et al. (2019), the authors introduce trapdoors to detect white-box adversarial examples attacks in order to trick the adversary's attack into converging towards specific adversarial directions, which later enables a straightforward detection. To this end, to defend a model against targeted adversarial examples towards label $y_t$, the clean data set of input-label pairs $(x, y)$ is augmented with *trapdoored* pairs $(x + \Delta, y_t)$, where $\Delta$ is a specific trigger. Then, the target model is trained by minimizing the cross-entropy between the clean and trapdoored input-label pairs to obtain the *trapdoored model*. An input $x$ is detected as a targeted adversarial example thanks to a threshold-based test, that compares a "*neuron activation signature*" of this input against that of the embedded trapdoors. Note that the authors extend their approach to untargeted adversarial examples. We refer to the article (Shan et al., 2019) for more details.

The authors mention that transferability between the target and the trapdoored model is very poor. Based on the provided code[3], we train trapdoored models with the architectures we used for $M$. We observe that the transferability is hight from $M$ to the trapdoor version but, on the contrary, the transferability is weak when adversarial examples are crafted on the trapdoored model and transferred to the target model. For the latter scenario, we report in Table 20 the worst $\mathrm{AC}_M$ and DAC values for FGSM, MIM, DIM, MIM-TI and DIM-TI along with those of our method. The parameters used to run the attacks are the same as described in Section 4. These results would correspond to a defense similar to ours where the public model is not the augmented model $M \circ P$, but the trapdoored model trained from $M$. The threat model stays the same, corresponding to an adversary which wants to transfer adversarial examples from the public model to the target model $M$. Results in Table 20 on MNIST and CIFAR10 for DAC indicate that using trapdoors could allow for a more efficient detection of adversarial examples, especially for large adversarial perturbations. Results for the $\mathrm{AC}_M$ values on the three data sets also corroborate the idea that both methods could benefit from each other.

Table 20: Optimal $\mathrm{AC}_{MoP}$, $\mathrm{AC}_M$ and DAC values (among FGSM, MIM, DIM, MIM-TI and DIM-TI) for the Trapdoor and Luring approaches.

|  | $\epsilon$ | TRAPDOOR | | | LURING | | |
|---|---|---|---|---|---|---|---|
|  |  | $\mathrm{AC}_{MoP}$ | $\mathrm{AC}_M$ | DAC | $\mathrm{AC}_{MoP}$ | $\mathrm{AC}_M$ | DAC |
| MNIST | 0.2 | 0.06 | 0.82 | **0.97** | 0.19 | **0.92** | 0.93 |
|  | 0.3 | 0.0 | 0.54 | **0.94** | 0.13 | **0.69** | 0.75 |
|  | 0.4 | 0.0 | 0.27 | **0.87** | 0.07 | **0.34** | 0.45 |
| SVHN | 0.03 | 0.1 | 0.48 | 0.58 | 0.11 | **0.81** | **0.87** |
|  | 0.06 | 0.07 | 0.38 | 0.48 | 0.0 | **0.58** | **0.71** |
|  | 0.08 | 0.01 | 0.23 | 0.38 | 0.0 | **0.48** | **0.67** |
| CIFAR10 | 0.02 | 0.0 | **0.71** | **0.84** | 0.03 | 0.58 | 0.64 |
|  | 0.03 | 0.0 | **0.58** | **0.77** | 0.0 | 0.30 | 0.39 |
|  | 0.04 | 0.0 | **0.45** | **0.78** | 0.0 | 0.16 | 0.27 |

---

[3]https://github.com/Shawn-Shan/dnnbackdoor

