# OpenReview forum: "Luring of transferable adversarial perturbations in the black-box paradigm"
_ICLR.cc/2021/Conference — Reject_

### Official Review · AnonReviewer2 · 2020-10-15
**A conceptually new framework for addressing the problem of adversaries in black box settings with promising results**

**Rating:** 8
**Confidence:** 3

**Review:**

Summary :

The paper proposes a new framework for addressing the problem of adversaries in black box settings in order to improve model robustness. Leveraging classical deception frameworks used in network security, the authors propose to fool the attacker by training what they call a `luring component’ that is augmented to an already trained model such that the new model does not later good samples and targets the adversaries to achieve the desired result. Additionally, the proposed framework does not need access to labeled data and can be applied to any pre-trained model. Promising results are demonstrated on multiple datasets like MNIST and CIFAR 10, etc.

Positives:

Novelty: the paper proposes a new framework which is distinct from existing methods that mostly consider white box settings, rely on robust features or anticipate perturbations. Instead, the proposed method is applicable to black box settings, is data agnostic, and can be applied to any pre trained model.

Technical sophistication:

1. The intuition behind the idea is explained well. Though the concept is borrowed from network security literature, its introduction and adaptation to the ML community is valuable.
2. The authors introduce two objectives that are designed towards training the luring component, and these are intuitive as well. In order to achieve their objective, the authors introduce a new loss function called the luring loss.
3. The algorithm for training the luring component also seems reasonable.

Experimentation:

1. Extensive comparison against baselines is detailed for characterization of luring effect as well as for metric evaluation.
2.  Results are reported on multiple datasets.

Presentation:
The paper is well motivated and fairly clear.

Overall comments:

The paper opens a new direction to adversarial machine learning research by proposing a new conceptual framework for addressing adversaries in black box settings. It offers practical value given that it is data agnostic , applicable to black box settings, and can be applied to any pre trained model.

---

> ### Author Response · Authors · 2020-11-18
> **Response to Reviewer#2**
>
> We thank the reviewer for this review, and particularly for having emphasized the novelty of our approach.

---

### Official Review · AnonReviewer4 · 2020-10-27
**ICLR 2021 Conference Review Paper822**

**Rating:** 6
**Confidence:** 4

**Review:**

Summary:
* This paper presents a novel method to prevent transferability of adversarial examples in black box settings. They name the effect “luring effect” which thwarts transferability of adversarial examples between two models. They evaluate on MNIST, SVHN, CIFAR10, and analyze the scalability on ImageNet.

Strengths:
* Thwarting transferability is a relevant and challenging research problem
* Relies only on logits, and can be applied to any pre-trained model
* Experimental comparison with related methods (Stack, Auto, CE)
* Code is publicly released.

Weaknesses:
* You evaluate robustness of using the luring effect as a defense with SPSA and ECO, but you do not consider an adaptive attacker that knows your defense and tries to attack it. See best practices discussed in [1].
* Related work section only mentions two closely related papers.
* Focus on black-box only may overestimate the effectiveness of the luring effect. It would be anyway interesting to know what happens in a white-box setting, or it if has any effect at all (see also guidelines in [1]).

Comments:

I think this paper tackles a very important problem and is generally very well-written, and I would like to thank the authors for releasing their code to the community, which definitely foster a faster advance of the state of the art. The proposed luring effect is based on an interesting idea, and is generally well-formalized and explained.

I do see some limitations in the current version of the paper and I would like to provide some recommendations that can improve its quality:
* The main limitation is somehow the lack of an adaptive attacker to evaluate the robustness of the luring effect. What if someone knows your luring effect and tries to bypass it? I think this type of reasoning is somehow missing from the paper. See also [1] for more details on what I am referring to.
* Real-world scenarios in which this may be useful are not fully clear to me. Let me explain a bit better. In your threat model, you assume that the attacker has access to infinite queries. Even if you introduce the “luring effect”, it is unclear to me what the attacker needs to achieve: there must be some obvious classifier feedback that the attacker may want to trigger, to evade the ML-based API. If the real underlying model M is not fooled, then the attacker may somehow know that his evasion is erroneous. However, if all replies and outputs are generated from model T, then in practice if the attacker evades T, from their perspective the attacker completed the evasion task successfully, even if the defender knows the result in M.
* Moreover, I would like to encourage the authors to clarify more intuitions about theoretical guarantees a user may have of not getting misleading decisions on vanilla (i.e., non-adversarial) classification requests after applying the luring effect. This was not entirely clear to me.
* As a minor final comment, I think it would be anyway useful to understand better if using a white-box threat model would actually make the luring effect meaningless (because it is relying on a gradient-free attack), and whether there are some gray-box scenarios in which there are still some properties for the luring effect.

References:
* [1] Carlini, Nicholas, et al. "On evaluating adversarial robustness." arXiv preprint arXiv:1902.06705 (2019). - “Live" paper on GitHub

---

> ### Author Response · Authors · 2020-11-18
> **Response to Reviewer#4**
>
> We thank the reviewer, notably for the insightful comments about an adaptive adversary and the extension to gray-box settings.
>
> **Threat model comments**
>
> We clarify the threat model that corresponds to an adversary crafting adversarial examples on the augmented protected model $T = M \circ P$ to transfer them to the target model $M$. Importantly, $M$ is completely hidden: the adversary does not have access to $M$ output.
> The luring effect leads to prediction differences that may be exploited into a classical adversarial example detection scenario. Moreover, because adversarial examples crafted from $T$ are likely to be unsuccessful on $M$, our approach is relevant for a transferability scenario. The general framework for transferability is the use of a surrogate model $M'$ to craft adversarial examples and transfer them to a target model $M$. By design, our approach aims at protecting deployed models that may be used as a surrogate models to strike $M$.
> For example, typical real-world use cases can be related to the widespread models deployment on a large variety of mobile devices. Current good practices are predominantly focused on SW and HW protections to keep models in a black-box flavor.
> A core model $M$ dedicated to a specific task (e.g. a biometric authentication process) is used in several systems and devices with different security environments and requirements. Simultaneously, clones or variations (i.e. $M$ or $M'$) of this model can be simultaneously deployed and used in more mainstream systems with looser security requirements (such as unlimited querying ability).
> In the hardware security, these real-world scenario (called profiling attacks) that put clone devices/systems at stake to strike more critical targets are major concerns, officially taken into account in the  international standard for computer security certification (CC [2] or FIPS 140 for cryptographic module [3] or upcoming AI certification [4]).
> By acting on these deployed models (deceiving the attacker) we strengthen the security of the critical target model $M$.
>
> **Adaptative adversary, white and gray-box settings**
>
> In our threat model, the supplementary knowledge given to an adaptive adversary is letting him know that $T$ has been trained with the luring loss. $M$ has to stay hidden, otherwise the adversary could use a decision-based attack ([5]).
> This being stated, an adaptive attacker knowing that $T$ has been trained with the luring loss grants the adversary no advantage to craft adversarial perturbations which transfer to $M$. Indeed, it would imply for the attacker to inverse the effect of the luring loss on the prediction output vector of $T$, which appears as a prohibitive effort.
>
> Our deceiving-based method is a pure black-box approach and is inefficient in a white-box setting since the attacker may simply extract $M$ from $T$ and then thwart it.
>
> However, we are convinced that our approach may be exploited with a more permissive threat model. In our opinion, the most viable solution would to be able to cause the luring effect between two models $M$ and $M'$ with different architectures. $M'$, trained with the luring loss, could be publicly released without restriction (in a white-box setting). The adversary could mount gradient-based attacks on $M'$ without being able to access $M$. The possible efficiency of such future work is backed by results with the MIM-W attacks, which corresponds to $T$ being attacked by gradient-based attacks (to anticipate future efficient attacks in the paper).
>
> **Impact on the clean examples**
>
> Let's say that the prediction given by $M$ corresponds to *"class A is predicted, class B is the second possible class"*. The prediction given by $T$ corresponds to *"A is predicted, the higher confidence given to A, the smaller confidence given to B"*. $M$ and $T$ share a common objective: A is the predicted class. It ensures that learned useful features by $M$ and $T$ correspond to the classification of clean examples in A. This point is notably demonstrated with experiments results reported in Table 8 (Appendix C), where $M$ and $T$ show minor disagreements for clean examples.
>
> We hope that we address your concerns with sufficient clarity. We follow your remarks in the revised version:
>
> **(1)** better emphasize on real-world scenarios corresponding to our threat model (Section 4.1.) For clarity, we change the naming of the scenario
>
> **(2)** include a discussion about an adaptive adversary and extensions of the luring effects in the white-box and gray-box settings (Section 5 and Appendix I)
>
> **(3)** better describe the intuition behind the luring effect (Section 2.4)
>
> References :
>
> [1] CC, ISO/IEC 15408,  https://www.commoncriteriaportal.org/
>
> [2] FIPS, ISO/IEC 17825,  https://csrc.nist.gov/publications/detail/fips/140/3/final
>
> [3] ISO/IEC TR 24028:2020 https://www.iso.org/standard/77608.html
>
> [4] Cheng et al. "Sign-OPT: A Query-Efficient Hard-label Adversarial Attack". ICLR 2018

---

> > ### Comment · AnonReviewer4 · 2020-11-23
> > **Feedback to response**
> >
> > Dear authors,
> >
> > thanks for your detailed answer.
> >
> > **Threat model**. I now appreciate better the use cases of your black-box threat model assumption, and what the luring attack can be useful for. Thanks for adding clarifications to the revised version of the paper, I think it greatly benefits the clarity of the work.
> >
> > **Adaptive adversary**. It is now clearer to me what are the assumptions of your adaptive attacker.
> >
> > **Overall**. Your answer compelled me to slightly update my score to 6. Nevertheless, I am still not fully convinced that the adaptive adversary model you are considering is the most appropriate one. The scenario you mention in your comment (which I report below) seems extremely interesting and relevant, and I would have liked to see it in the current version of the paper as well, to demonstrate some additional robustness property of the luring effect in a white/gray box setting:
> >
> > > [...] However, we are convinced that our approach may be exploited with a more permissive threat model. In our opinion, the most viable solution would to be able to cause the luring effect between two models  and  with different architectures. , trained with the luring loss, could be publicly released without restriction (in a white-box setting). The adversary could mount gradient-based attacks on  without being able to access . The possible efficiency of such future work is backed by results with the MIM-W attacks, which corresponds to  being attacked by gradient-based attacks (to anticipate future efficient attacks in the paper).

---

> > > ### Author Response · Authors · 2020-11-24
> > > **Response**
> > >
> > > We want to thank you for updating the review. We updated the paper by moving the comment you report about the gray-box setting from the Appendix I to the Discussion and Related Work section (5).

---

### Official Review · AnonReviewer1 · 2020-10-30
**Concerns about threat model, baselines**

**Rating:** 5
**Confidence:** 2

**Review:**

**Update:**

Thanks to the authors for their detailed response to my review. Unfortunately after reading the response, I don't understand how it addresses some significant concerns I have about this paper and therefore I can't increase my score. In particular:

- Author response says "Measuring the transferability from $M$ to $M_{adv}$ is actually the opposite of our concern, since our threat model deals with the transferability from $T$ (the protected augmented model) to $M$ (the target model)." I'm suggesting defining $T$ as a vanilla model and $M$ as a PGD-trained model. I don't see why your threat model would preclude such a choice.
- Author response says "$M$ is hidden from the adversary. Then, the adversary does not have access to $M$'s output, and cannot mount a black-box attack to thwart a binary classifier $M(x)/T(x)$." The adversary doesn't need full access to $M$, only access to the output of the binary classifier $M(x)/T(x)$. Presumably they have at least indirect access to the binary prediction; otherwise the defender would be able to detect adversarial inputs but wouldn't be able to take any meaningful action based on the detection  (e.g. denying access to the attacking user, etc...).

There is still a possibility that I've misunderstood something fundamental, so I leave my review confidence as fairly low.

(Original review below)
---

Summary:

The authors propose a threat model for adversarial robustness and a corresponding algorithm to defend against attacks in this model.
The threat model supposes that attackers have unlimited black-box access to one model and limited access to a second model which is the target of the attack.
The algorithm trains a model $M \circ P$ and a second model $M$ such that they agree on clean data but disagree on adversarial examples (the authors call this "luring"; i.e. $M \circ P$ "lures" adversaries into producing adversarial examples which don't work on $M$).

Review:

I'll disclaim that I don't know very much about adversarial robustness, so I've assigned a low confidence to my review.
With that said, I have concerns regarding the threat model described in the work (namely, that it seems artificial) and the lack of baselines (see "correctness" below) which prevent me from recommending acceptance at this time.

Correctness:

The experiments indeed show that the "luring behavior" exists; i.e. that attacks on $M \circ P$ don't transfer to $M$ as easily when $M$ and $M \circ P$ are trained using the luring objective as when they're trained using other baseline strategies.

There's a natural baseline which seems to be missing: we can train two networks independently from scratch, one with PGD and one without. If the adversarial examples from the vanilla-trained model don't transfer to the PGD-trained model, then there's no need for a special objective or removable neural net component. (This missing baseline is especially conspicuous given the experiments on PGD-trained models in Appendix H.2).

Novelty:

The method proposed is novel to my knowledge, but I disclaim that I'm not familiar with adversarial defense literature.

Significance:

My main concern with this paper is with regard to the threat model.
The authors give two possible scenarios for the defender. In the first ("inner transferability scenario"), we assume $\mathbb{S}_D = \mathbb{S}_A$ and the defender detects adversarial inputs by comparing $M(x)$ and $(M \circ P)(x)$.
But this amounts to doing binary classification on the inputs, and the attacker can perform a black-box attack on this binary classifier to generate adversarial examples which fool both $M$ and $M \circ P$.

In the second scenario ("distant transferability"), we assume that $\mathbb{S}_D$ is secured (i.e. the attacker doesn't have access). Under this model the proposed approach makes sense, but the threat model seems contrived. In what situations are we able to limit access to a model, and yet required to provide access to a second model which makes the same predictions on clean data? The paper would be much stronger if the authors could motivate this threat model by real-world examples (or even hypothetical ones).

Clarity:

Overall the paper reads acceptably clearly. I appreciate that the authors provide a clear description of their threat model (notwithstanding my concerns with that model). From what I know of adversarial example papers, this is important and often overlooked.

This is subjective, but I feel the result tables are hard to interpret at times because of heavy use of acronyms (e.g. $DAC$, $AC_M$, $C_E$, etc...) where plain-text would suffice, and also because many of the rows/columns aren't labeled (e.g. in Table 1, what type of things are "SPSA", "ECO", "MIM-W"? What type of things are "STACK", "AUTO", "C_E", "LURING"? Don't assume readers are familiar with these acronyms, nor that they've read the full text before reading the tables.)

---

> ### Author Response · Authors · 2020-11-18
> **Response to Reviewer#1**
>
> We thank the reviewer for these comments,  which significantly help us to improve the quality of our paper.
>
> **On the Correctness remark**
>
> You suggest a baseline by training independently, one model with Adversarial Training [1] and the other with classical training.
> The baselines in Section 3 are first defined for characterization purpose since we need to check if other architectures (following the same principle of an additional component $P$) may also induce the objectives of our deceiving approach, i.e.*prediction neutrality* and *adversarial luring*.
> \
> Your baseline concerns the evaluation of the adversarial robustness (Section 4). However, this baseline cannot not match our threat model. Let's note $M$ the vanilla model and $M_{adv}$ the PGD-trained model. Measuring the transferability from $M$ to $M_{adv}$ is actually the opposite of our concern, since our threat model deals with the transferability from $T$ (the protected augmented model) to $M$ (the target model).
> Moreover, considering the transfer scenario from $M_{adv}$ to $M$, nothing protects $M$ from successful adversarial examples crafted on $M_{adv}$.
> That explains why we do not consider the Adversarial Training as a concurrent baseline but as a complementary protection. We are deeply convinced that our deceiving approach (relevant in specific threat models) and other defenses can be fruitfully combined to answering a wider attack surface.
>
> **On the Significance remarks.**
>
> Concerning the first scenario, that represents a pure adversarial example detection. The attacker queries T and crafts adversarial examples thanks to $(x, T(x))$ but the intrinsic comparison of $M(x)$ and $T(x)$ enables to thwart the attack process. The adversary can query without limitation the augmented model $T$ and get logits outputs, but, importantly, $M$ is hidden from the adversary. Then, the adversary does not have access to $M$'s output, and cannot mount a black-box attack to thwart a binary classifier $M(x) / T(x)$.
>
> The second scenario exploits the fact that most adversarial examples crafted from $T$ will be unsuccessful on $M$. We remind that the general framework for transferability is an adversary using a surrogate model $M'$ to craft adversarial examples and transfer them to a target model $M$. Note that depending on the context (task, adversary's knowledge...), $M'$ is already at his disposal or he needs to build it. By design, our approach aims at protecting deployed models that may be used as a surrogate models to strike a critical target model $M$.
>
> For that scenario, typical real-world use cases can be related, among others, to the widespread models deployment on a large variety of mobile devices (e.g. the so-called 'edge AI' for the IoT domain).
> As for cryptographic modules, current good practices for edge neural networks are predominantly focused on the use of software and hardware protections to keep models in a black-box flavor [1].
> A core model $M$ dedicated to a specific task (e.g. a biometric authentication process) is used in several systems and devices with different security environments and requirements. Simultaneously, clones or variations (i.e. $M$ or $M'$) of this model can be simultaneously deployed and used in more mainstream systems with looser security requirements (such as unlimited querying ability).
>
> We highlight the fact that, in the hardware security, these real-world scenario (called profiling attacks) that put clone devices/systems at stake to strike more critical targets are major concerns, officially taken into account in the  international standard for computer security certification (CC [3] or FIPS 140 for cryptographic module [4] or ISO/IEC TR 24028:2020 for Artificial Intelligence upcoming certification [5]).
> By acting on these deployed models (that is, by fooling potential attacker) we strengthen the security of the critical target model M.
>
> We hope that we address your concerns with sufficient clarity. In the revised version of the paper, we performed the following modifications:
>
> **(1)** better link the threat model with real-world scenarios (Section 4.1) and, for clarity, we change the naming of the scenario
>
> **(2)** detail the complementarity of our method with other existing defense methods, such as Adversarial Training (Section 4.4 second paragraph)
>
> **(3)** better detail some acronyms in the Tables to improve the readability
>
> References:
>
> [1] P. M. VanNostrand, et al., Confidential Deep Learning: Executing Proprietary Models on Untrusted Devices, eprint 1908.10730, 2019
>
> [2] A. Madry, et al. Towards deep learning models resistant to adversarial attacks. ICLR 2018
>
> [3] CC, ISO/IEC 15408,  https://www.commoncriteriaportal.org/
>
> [4] FIPS, ISO/IEC 17825,  https://csrc.nist.gov/publications/detail/fips/140/3/final
>
> [5] ISO/IEC TR 24028:2020 https://www.iso.org/standard/77608.html

---

### Official Review · AnonReviewer3 · 2020-10-31
**Official Blind Review #3**

**Rating:** 5
**Confidence:** 4

**Review:**

In this paper, the authors present a novel approach to evade the transferability of adversarial examples between two models. Specifically, they design a luring loss to train model T, an augmented version of M, where the adversarial examples cannot transfer from T to M. The luring loss is designed to reach a twofold objective: (1) for a clean example, T and M yield the same prediction (2) for an adversarial example, T and M yield different predictions, and in the best case, M can provide correct prediction. The proposed approach can serve as a defense for both detect adversarial examples and defend adversarial examples. Experimental results show that the proposed defense can detect and defend adversarial examples better that the three baselines. In general, the paper is clearly written and easy to follow.

One weakness of this paper is the reason of why luring loss can make the non-robust feature of T different from M is not sufficiently explained.

Another weakness is that, in the setting of black-box paradigm, the authors consider both adversarial detection and defense, but they did not use recent detection or defense baselines. This makes their experimental results less convincing.

---

> ### Author Response · Authors · 2020-11-18
> **Response to Reviewer#3**
>
> We thank the reviewer for this careful reading and take these comments into account to improve and clarify our paper. We propose the following point-to-point response to the two highlighted concerns.
>
> **1)** We bring clarifications to the intuition behind the luring effect in terms of features learned (by the target and the augmented models).
> Let's formulate the prediction of $M$ as: *"class A is predicted, class B is the second possible class"*. To reach this prediction, M learned some "useful features" (following the terminology of the framework from Ilyas et al. [1]). After the training of $P$ with the luring loss, the prediction given by $T = M \circ P$ becomes *"class A is predicted, the higher confidence given to class A, the smaller confidence given to class B"*. This conceptualization of the two predictions (given by $M$ and $T$) brings to the intuition behind the luring effect: since concepts learned by $M$ and $T$ are essentially different, $M$ and $T$ rely on different useful features (even if they still share the same objective, i.e. A is the output), and therefore the two models display a different (but fruitful) type of sensitivity to an input perturbation.
> The luring effect causes two possible and desired effects: either (in the best case) useful features learned by $M$ will be resilient to perturbations fooling $T$ (i.e. will be robust to these perturbations), or will respond differently to these perturbations (given two different adversarial labels).
>
> This intuition, which is the core basis of the luring effect, is supported afterwards by experiments:
>
> **i.** Since MNIST has a uniform background, it facilitates the visualization of $P$'s impact. We observe significantly more pixels modified by adversarial perturbations with the luring effect, than the other baselines used for characterization. Moreover, saliency maps show that $M$ and $T$ rely on clearly different features. This agrees with the intuition that non-robust useful features require different types of perturbations to be "flipped" in the same way. Experiments are detailed in Section 3.3, and visualizations of these results in Figure 3 (left and right).
>
> **ii.** For SVHN and CIFAR10, the same pixel modification makes logits of $M$ and $T$ vary in different directions way more with the luring effect than the baselines used for comparison. Experiments are detailed in Appendix D.3.
>
> **2)** Secondly, we address your remark related to existing methods. To our knowledge, no other method relies on our deceiving approach except [2] (see Section 5 and Appendix I) but in a pure white-box setting. First, black-box defense is under-represented and it has to be noted that there is very little work on thwarting transfer black-box attacks in particular (it is also the case with defenses against black-box query-based attacks). Moreover, the existing detection and defense methods are in fact complementary to ours, in the threat model we consider: an adversary in the black-box setting, that exploits the transferability property to transfer adversarial perturbations from the model $T$ to the target model $M$.
> Indeed, in this threat model, the luring effect can be combined with existing schemes, as its principle is conceptually different from them, and thus thwarts the adversary at different levels.
> Another defense method, this time focusing on $M$ and designed to protect $M$, can totally be added to the luring effect to better thwart the adversary. This method can be either a reactive method (purification or detection), or a proactive one, such as Adversarial Training [3], as outlined in Section 4.4 (with details in Appendix H.2).
>
>
> We thank you for your remarks and we hope that the modifications of the revised paper (listed below) correspond to your expectations.
> The modifications in the revised paper are as follows:
>
> **(1)** better description of the intuition behind the luring effect (Section 2.4, first paragraph) in terms of useful features
>
> **(2)** emphasize on the complementarity of our scheme with existing schemes (Section 4.4, second paragraph)
>
>
> References:
>
> [1] A. Ilyas, et al. Adversarial examples are not bugs, they are features. NIPS 2019
>
> [2] S. Shan, et al. Using honeypots to catch adversarial attacks on neural networks. ACM CCS 2020
>
> [3] A.  Madry, et al. Towards deep learning models resistant to adversarial attacks. ICLR 2018

---

### Public Comment · ~Nicholas_Carlini1 · 2020-11-12
**Gradient masking effects?**

Figure 1 appears to show that luring is effective at epsilon=0.4 on MNIST, with a success rate above 90%. What is the accuracy an adversary can reduce the model to with a distortion of 0.5? If it is any greater than 10%, then this would indicate that the model is masking gradient information.

---

> ### Author Response · Authors · 2020-11-18
> **Answer to gradient masking concerns**
>
> Thank you for commenting our paper.
>
> We remind that our work sets in the black-box paradigm with a threat model corresponding to an adversary who crafts adversarial examples on the augmented model $T = M \circ P$ to transfer them to the target model $M$.
>
> Figure 2 does not correspond to an accuracy measure but two metrics focused on our twofold objective: the proportion of successful adversarial examples on $T$ for which $T$ and $M$ disagree (noted DR) and the proportion of successful adversarial examples on $T$, which $M$ classifies correctly (noted IAR).
>
> We confirm that even for epsilon=0.4, the gradient-based PGD attack can successfully reduce the adversarial accuracy to 0 for the model $T = M \circ P$. This is also naturally the case for the target model $M$, which is trained classically with the cross-entropy loss.
>
> You are perfectly right to highlight the 'gradient masking' problem. Following guidelines presented in [1], we had previously checked that the luring effect did not induce gradient masking for the model $T = M \circ P$ when attacked in a white-box setting (as noted in Section 4.3). To this end, we performed the following sanity checks:
>
> - increasing the perturbation budget does decrease the adversarial accuracy
>
> - the gradient-free SPSA attack does not perform better than the gradient-based PGD attack
>
> - the robustness against black-box transfer attacks from  a  model  with  the  same  architecture  does  not perform better than PGD.
>
>
> References:
>
> [1] Carlini, Nicholas, et al. On evaluating adversarial robustness. arXiv preprint arXiv:1902.06705 (2019).

---

### Author Response · Authors · 2020-11-18
**General response**

We thank the reviewers for their very constructive comments, which help us update our paper with a gain of clarity. We bring a specific and detailed answer for each review below.

We remind the logical flow of our work:

(i) *main idea* : fooling the attacker with wrong adversarial direction (lures) by acting on the logits sequence order.

(ii) *intuition* : using the features-based framework (from Ilyas et al. [1]), make the luring model $T = M \circ P$ and the target model $M$ rely on useful features which behave differently when facing adversarial perturbations.

(iii) *characterization and evaluation* : experimentally isolate the luring effect from potential other factors and evaluate the relevance of our approach in term of black-box robustness.


The update of the paper aims at clarifying our approach and better show the originality of our work.
To summarize:

**(1)** We bring supplementary clarifications for the threat model by pointing out real-world use-cases (in Section 4.1)

**(2)** We detail the intuition behind the luring effect in terms of robust and non-robust useful features (in Section 2.4)

**(3)** We emphasize on the conceptual novelty of the luring effect, to demonstrate that it can be used in combination with
          other defense methods (notably Adversarial Training) to protect a model (in Section 4.4)

**(4)** We discuss the case of an adaptive adversary and the extension of the luring effect in a more permissive threat model
          (in Section 5).


We hope that these updates meet your expectations and we remain at your disposal for any other precision and enhancements that could enable the acceptance of our work.


References:

[1] A. Ilyas, et al. Adversarial examples are not bugs, they are features. NIPS 2019

---

### Decision · Program_Chairs · 2021-01-07
**Final Decision**

**Decision:**

Reject

**Comment:**

The paper proposes to augment the original  model to introduce the "luring effect", which can be used for detection and black-box defense. Despite being an interesting setup, there are several weaknesses in the threat model (whether it is practical) and the evaluation (lack of adaptive attacks). Those concerns remain after the rebuttal phase.

Threat model: see the concerns raised by Reviewer 1 and the updated comments after the rebuttal phase.

Lack of adaptive attack: the authors assume that the attacker has very limited knowledge to how the system works. This could be viewed as a "black-box setting" for adversarial detection evaluation, and actually many other detection works can perform almost perfectly in this setting, so it's not clear how significant are the results. The authors tried to consider some adaptive attacks in their rebuttal but the reviewers are still not fully convinced.